# Genotypic Diversity of *Candida parapsilosis* Complex in Invasive Candidiasis at a Pediatric Tertiary Hospital: A 5-Year Retrospective Study

**DOI:** 10.3390/jof8121280

**Published:** 2022-12-06

**Authors:** Luiza S. Rodrigues, Adriele C. Siqueira, Regiane N. Spalanzani, Thaís M. Vasconcelos, Bianca Sestren, Saloe P. Bispo, Renata B. V. Abreu, Letícia Kraft, Marinei C. Ricieri, Fábio A. Motta, Libera M. Dalla-Costa

**Affiliations:** 1Instituto de Pesquisa Pelé Pequeno Príncipe, Av. Silva Jardim, 1632-Água Verde, Curitiba CEP 80250-060, PR, Brazil; 2Hospital Pequeno Príncipe, Desembargador Motta, 1070-Água Verde, Curitiba CEP 80250-060, PR, Brazil; 3Coordenação-Geral de Laboratórios de Saúde Pública, do Departamento de Articulação Estratégica de Vigilância Em Saúde, da Secertária de Vigilância em Saúde do Ministério da Saúde, SRTVN 701, Via W5 Norte, Brasília CEP 70719-040, DF, Brazil

**Keywords:** invasive candidiasis, *Candida parapsilosis* complex, pediatric, biofilm, microsatellite

## Abstract

Invasive candidiasis (IC) contributes to the morbidity and mortality of hospitalized patients and represents a significant burden to the healthcare system. Previous Brazilian studies have reported the presence of endemic *Candida parapsilosis* sensu stricto genotypes causing candidemia and clonal transmission involving fluconazole-resistant isolates. We performed a 5-year retrospective analysis of IC cases in a Brazilian tertiary pediatric hospital and conducted a molecular investigation of *C. parapsilosis* sensu stricto. Non-duplicate *C. parapsilosis* sensu stricto genotyping was performed by microsatellite analysis. Antifungal susceptibility and biofilm formation were also evaluated. A total of 123 IC episodes were identified, with an IC incidence of 1.24 cases per 1000 hospital admissions and an overall mortality of 34%. The main species were the *C. parapsilosis* complex (35.8%), *Candida albicans* (29.2%), and *Candida tropicalis* (21.9%). All *C. parapsilosis* sensu stricto were recovered from blood cultures, and 97.5% were biofilm producers. Microsatellite typing identified high genotypic diversity among the isolates. We observed that all isolates were sensitive to amphotericin B, and although one isolate was non-sensitive to fluconazole, only a silent mutation on ERG11 gene was identified. No clear evidence of clonal outbreak or emergence of fluconazole-resistant isolates was found, suggesting that multiple sources may be involved in the epidemiology of IC in children.

## 1. Introduction

Healthcare-setting-associated infections are important causes of mortality and increased medical costs among hospitalized patients worldwide. Invasive candidiasis (IC) is the main fungal disease among inpatients, with a high incidence in South America [1]. Although it is generally an endogenous infection after prior colonization of the gastrointestinal tract or skin, the disease can also be caused by exogenous strains acquired during a hospital stay [2]. Despite *Candida albicans* being the major cause of IC worldwide, species of the *Candida parapsilosis* complex (*Candida parapsilosis*, *Candida metapsilosis*, and *Candida orthopsilosis*) are generally the most common non-albicans *Candida* species, especially in newborn infants and pediatric patients [1,3,4,5,6,7]. Clinically, *C. parapsilosis* sensu stricto is the main species among the cryptic complex species [7]. The species can grow in parenteral nutrition, form biofilms on medical devices, and persist in the hospital environment, which are attributes that facilitate its spread in hospital environments [2].

Recently, the World Health Organization (WHO) published a list of priority fungal pathogens in which *C. parapsilosis* is in the high-priority group, associated with mortality ranging from 20 to 45% in invasive disease despite active antifungal treatment; this report highlights, among other concerns, the lack of systematic surveillance of this opportunistic pathogen [8]. Therefore, it is important to consider that some aspects of pediatric and adult IC significantly differ, such as host factors, pharmacokinetics, and outcomes that underscore the need for specific research focusing on pediatric patients.

Outbreaks involving *C. parapsilosis* sensu stricto have been reported in diverse geographical regions, and some of them exhibited evidence of horizontal transmission through healthcare workers’ hands or intensive care unit (ICU) surfaces [9,10,11,12,13]. Previous Brazilian studies have reported the presence of endemic genotypes causing candidemia and clonal transmission involving fluconazole-resistant *C. parapsilosis* sensu stricto strains (FRCP) associated with high mortality rates [11,14,15,16]. In this context, surveillance is important to determine the epidemiology of this species complex, observe changes in species distribution, and detect resistant strains, especially in pediatric environments. Answering these questions in pediatrics will provide important epidemiological knowledge for this age group in which the occurrence of *C. parapsilosis* sensu stricto is prevalent.

Studying the genetic relationship between *Candida* spp. isolates from patients and hospital environments is important because it may uncover the presence of endemic genotypes, which suggests a common reservoir or horizontal transmission [17,18,19,20]. Microsatellite analysis has been used as one of the most common typing tools with high discriminatory power and reproducibility, as it is able to identify specific genotypes and the genetic relationship between strains. This typing method can, therefore, distinguish clonal clusters from genetically unrelated genotypes and has proven to be a valuable tool for supporting epidemiological investigations [14,17,18,19,20,21,22]. To better understand hospital *C. parapsilosis* sensu stricto epidemiology in pediatric patients and determine whether its occurrence is associated with clonal outbreaks and emergence of fluconazole-resistant isolates in adult patients, we conducted a 5-year retrospective study of IC cases with genotypic diversity analysis, susceptibility profiling, and biofilm formation characterization of *C. parapsilosis* sensu stricto isolates.

## 2. Materials and Methods

### 2.1. Background, Setting, and Isolates

We recovered clinical isolates from pediatric patients (between 0 and 18 years of age) with clinical and laboratory diagnoses of IC at a 372-bed pediatric tertiary-care teaching hospital in the south of Brazil between August 2016 and August 2021. Clinical samples (blood and other sterile body fluids) from patients with invasive infections were collected by appropriate aseptic procedures and sent to the microbiology laboratory for culture. Positive cultures for yeast were identified using matrix-assisted laser desorption ionization mass spectrometry (MALDI-TOF MS) with a MicroflexTM LT instrument (Bruker Daltonics, Billerica, MA, USA), according to the manufacturer’s instructions. The isolates were stored in skim milk and frozen at −80 °C until processing for further study. A total of 123 non-duplicate *Candida* spp. isolates were included in this study. Before testing, all isolates were cultured on Sabouraud dextrose agar (Neogen Corporation, Lansing, MI, USA) and chromogenic Candida agar (Laborclin, Pinhais, PR, Brazil) to ensure their viability and purity. The identification confirmation of all clinical isolates was performed by MALDI-TOF MS. Demographic and clinical data, including hospital unit, site of infection, and outcome, were collected from the medical records of all patients. The Institutional Review Board (IRB) of the participating center (IRB #2.096.359) approved this study. Investigations were carried out by securing each patient’s anonymity.

### 2.2. Biofilm Assay

Total biofilm biomass was quantified by crystal violet staining in a 96-well microtiter plate as previously described [23]. Isolates were grown on Sabouraud dextrose agar (Merck, Darmstadt, Germany) for 24 h at 35 ± 2 °C. One colony from each plate was inoculated into 10 mL Sabouraud dextrose broth (Laboratorios Conda S.A., Madrid, Spain) for 18 h at 35 ± 2 °C under agitation at 120 rpm. Following incubation, cells were harvested via centrifugation at 3000× *g* for 10 min at 4 °C and washed twice with phosphate-buffered saline (PBS, pH = 7.5). Pellets were then suspended in RPMI 1640 (Sigma-Aldrich, St. Louis, MO, USA), and the cellular density was adjusted to 1–5 × 10^6^ cells/mL (0.5 McFarland). A total of 200 µL of the suspension was placed in each well of a 96-well, flat-bottomed microtiter plate and incubated for 24 h at 35 ± 2 °C under agitation at 120 rpm. The suspensions were discarded, and non-adherent cells were removed by washing the biofilms three times with PBS. Then, biofilms were fixed with 200 µL of methanol, which was removed after 15 min. The microtiter plates were allowed to dry at room temperature, and 200 µL of CV 1% *v*/*v* (Laborclin, Pinhais, Brazil) was added to each well and incubated for an additional 5 min at room temperature. The wells were gently washed three times with sterile ultra-pure water, and 200 µL of acetic acid (33% *v*/*v*) was added to release and dissolve the stain. The suspension (200 µL) was transferred to clean wells, and absorbance was measured in duplicated using Epoch Microplate Spectrophotometer (Agilent, Santa Clara, CA, USA) at 570 nm (OD570). Each experiment was performed in duplicate, including the reference *C. parapsilosis* ATCC 22,019 strain and sterile ultra-pure water that was used as a negative control. The optical density cut-off point (ODc) for biofilm formation was determined to be above the absorbance of the negative controls. The isolates were classified into four groups comparing the OD_570_ values of isolates and ODc: non-biofilm producer, OD ≤ ODc; weak biofilm producer, ODc < OD ≤ (2 × ODc); moderate biofilm producer, (2 × ODc) < OD ≤ (4 × ODc); and strong biofilm producer, (4 × ODc) < OD.

### 2.3. Antifungal Susceptibility Testing

In vitro antifungal susceptibility to amphotericin B (0.002–32 μg/mL) and fluconazole (0.016–256 μg/mL) was tested using gradient diffusion strips (all from bioMérieux, Marcy-l’Etoile, France) following the manufacturer’s instructions. Yeast inoculum suspensions were prepared as described by the Clinical and Laboratory Standards Institute (CLSI) M27-A3 [24]. Briefly, inoculum suspensions of *Candida* spp. isolates were prepared to a final concentration of 1–5 × 10^6^ cells/mL (0.5 McFarland). This suspension was used directly to inoculate RPMI-1640 agar plates containing 20 g/L D-glucose (Sigma-Aldrich, St. Louis, MO, USA). The inoculated agar surface was allowed to dry for 15 min before the gradient diffusion strips were placed on it. Thereafter, the plates were incubated at 35 °C for 24 and 48 h (if insufficient growth was present after 24 h). The assay was validated using the quality control isolates *Candida krusei* ATCC 6258 and *C. parapsilosis* ATCC 22 019. The final minimum inhibitory concentration (MIC) values were based on the consensus between two readers and interpreted according to current CLSI M60 clinical breakpoints [25]. Considering that the interpretation of the MIC value is dependent upon the antifungal drug class, for polyenes (amphotericin B), the MIC was interpreted as the value where there was 100% growth inhibition; and for azole, the MIC was interpreted as the value where there was 80% growth inhibition. CLSI has not determined breakpoints for amphotericin B; therefore, in this study, resistant isolates were defined as isolates with a MIC > 1 µg/mL.

### 2.4. Microsatellite Typing

Genotyping of all *C. parapsilosis* sensu stricto isolates and the reference *C. parapsilosis* ATCC 22,019 was performed using PCR amplification of eight polymorphic microsatellite markers as described by [22]. PCR products were separated on 3% agarose gel and visualized under UV illumination and on 18% acrylamide (29:1 acrylamide:bis-acryl-amide) gel and visualized following silver staining. The strains showing two PCR products in a microsatellite marker were typed as heterozygous, while those presenting a single amplification product were considered homozygous. The total allelic composition was then converted to binary data by scoring the presence “1” or absence “0” of each allele. Genotypes showing the same alleles for all eight markers were considered identical. Endemic genotypes were defined as identical genotypes infecting ≥ 2 different patients, and a cluster was defined as a group of ≥2 patients infected by an endemic genotype [17]. Endemic genotypes were confirmed after running the isolates in duplicate, with the patients involved in a cluster being geographically and temporally related.

### 2.5. Clustering Analysis

Data from microsatellite typing were treated as categorical, and the genetic relationships between the genotypes were determined in the R statistical computing environment (https://www.r-project.org/, accessed 27 October 2022, R Foundation for Statistical Computing, Vienna, Austria). The main R packages used were ComplexHeatmap, ggplot2.tidyverse, and cluster [26,27,28]. The UPGMA (unweighted pair group method with arithmetic mean method) tree produced from Bruvo’s distance was used for clustering [29]. The minimum coverage network using a distance matrix was developed by Poppr: an R package [29].

### 2.6. ERG11 Gene Sequencing

PCR amplification and sequencing of the *ERG11* gene—encoding lanosterol 14 α-demethylase—of the FRCP isolate were carried out using previously described specific primers [30]. PCR products were purified using ExoSAP-IT™ (Thermo Fisher Scientific, Waltham, MA, USA) and sequenced using a 3500 Genetic Analyzer (Applied Biosystems, Foster City, CA, USA). After curation of the sequences, they were aligned with the available wild-type corresponding sequence of *C. parapsilosis* ATCC 22,019 (GenBank accession no: GQ302972).

## 3. Results

A total of 123 IC episodes were identified in this 5-year retrospective analysis. The patients had a median age of 1 ± SD 4.8 (range 0–17 years old), among which 50.4% of patients were male. The overall annual IC incidence was 1.24 cases per 1000 hospital admissions, with an overall mortality of 34%. The main species identified were the *C. parapsilosis* complex (35.8%), followed by *C. albicans* (29.2%) and *Candida tropicalis* (21.9%) (Figure 1). Among the *C. parapsilosis* complex isolates (*n* = 44), 95.5% were *C. parapsilosis* sensu stricto, and only one *C. orthopsilosis* and *C. metapsilosis* were identified. All *C. parapsilosis* complex isolates were recovered from blood cultures. From a total of 42 *C. parapsilosis* sensu stricto isolates stored in skim milk and frozen at −80 °C, 40 survived preservation and were available for biofilm and antifungal susceptibility testing and microsatellite typing. Only one isolate was identified as a biofilm non-producer, and 75% of the total isolates were identified as strong producers (Figure 2). The results of antifungal susceptibility testing showed that all *C. parapsilosis* sensu stricto isolates were sensitive to amphotericin B, and one was non-sensitive to fluconazole, with a MIC of 6.0 µg/mL. The overall MIC_50_ and MIC_90_ for amphotericin B and fluconazole were 0.5 and 1 μg/mL, respectively (Figure 3). The *ERG11* gene of the isolate that was non-sensitive to fluconazole was amplified, sequenced, and compared with the reference ATCC 22,019 *ERG11* gene sequence. Only one silent mutation was identified (T591C).

Eight polymorphic microsatellite markers were used to type the 40 non-duplicate *C. parapsilosis* sensu stricto identified in the study to evaluate the similarities among the isolates and determine whether these isolates represented a persistent clonal lineage or unrelated strains. According to the microsatellite analysis, 38 different allelic profiles (genotypes) were assigned. Three isolates (7.5%) shared an identical allelic profile (endemic clone); therefore, a cluster with three patients was identified. The allelic profile of all isolates and *C. parapsilosis* ATCC 22,019 identified on microsatellite typing is presented in Figure 4, while Table 1 lists the general clonal isolates findings (*n* = 3). The relationships between 38 genotypes and the genotype distribution by hospital setting and year of isolation are shown in Figure 5.

## 4. Discussion

This study identified a total of 123 IC episodes, with an overall annual IC incidence of 1.24 cases per 1000 hospital admissions. A previous study reviewing candidemia (the most common form of IC) in South America reported an incidence range of 0.76–6.0 per 1000 hospital admissions [1]. In the present study, the *C. parapsilosis* complex was the major cause of IC, followed by *C. albicans* and *C. tropicalis*. In the same hospital, from 2008 to 2011, *C. albicans* was the main cause of candidemia, followed by the *C. parapsilosis* complex and *C. tropicalis*. However, between 2014 and 2017, an increasing prevalence of the *C. parapsilosis* complex in IC cases was already observed [3,31]. This *Candida* species distribution is consistent with previously published studies [15,32,33,34]. Even in centers where *C. albicans* is the major species involved in IC cases, the *C. parapsilosis* complex is often the main non-albicans species, especially in patients aged ≤ 2 years [1,4,7]. The distribution of *Candida* spp. in IC is an important factor since some species are associated with better outcomes than others. In adult and pediatric patients, *C. parapsilosis* complex candidemia is associated with lower mortality rates compared with *C. albicans* [1]. Among the *C. parapsilosis* complex isolates identified in this study, only one of *C. orthopsilosis* and *C. metapsilosis* were detected; *C. parapsilosis* sensu stricto has been the most reported member of the *C. parapsilosis* complex [32]. Biofilm is another concern associated with *Candida* spp. infections although *Candida* species differ in terms of biofilm formation. In all cases, the biofilm compromises antifungal treatment by limiting the penetration of substances through the extracellular matrix and protecting cells from host immune responses. If it occurred in implanted medical devices, implant replacement would often be required [23]. Moreover, biofilm formation has been previously suggested as risk factor for IC-related death in pediatric patients [3]. Here, all isolates were obtained from blood cultures, of which 97.5% were classified as biofilm formers.

Once the *C. parapsilosis* complex is recognized as an opportunistic pathogen that causes hospital-associated outbreaks, identifying the source of IC is important because of its implications for preventive strategies [12,33,35]. A previous study indicated that the *C. parapsilosis* complex and *C. albicans* expand in the gastrointestinal tract and can translocate into the bloodstream [36]. Cutaneous-originating candidemia has also been suggested in *C. parapsilosis* complex infections in the context of catheter-related candidemia [35]. Thus, the epidemiology of nosocomial *C. parapsilosis* complex infection is still undefined and may involve various origins including endogenous sources and the hospital environment [32]. Transmission of the *C. parapsilosis* complex from the hands of healthcare workers to neonates and children has been suggested as a contributing factor to the high occurrence of the species in pediatric populations [9,18,33]. In this context, isolate genotyping allows us to understand the role of the nosocomial transmission of *C. parapsilosis* stricto sensu strains in hospitalized patients. Herein, we used microsatellite typing, as it is a powerful tool to study the genetic relationship between *C. parapsilosis* sensu stricto isolates [9,11,13,14,15,16,17,22].

Here, a total of 38 genotypes were found among the studied 40 non-redundant isolates (Figure 4). Figure 5 shows the genetic relationship according to different hospital settings and year of isolation, where most genotypes were sporadically distributed. The diversity pattern of our findings suggests that multiple sources may be involved in IC occurrence and the likelihood of its endogenous origin, which does not minimize the fundamental role of hand hygiene in preventing infection. In line with our results, another Brazilian study, which investigated 19 cases of breakthrough candidemia in pediatric patients with a high occurrence of *C. parapsilosis* using microsatellite analysis, also did not identify evidence of horizontal transmission [15]. This finding was recently supported by another study that did not identify clear evidence of a clonal outbreak causing a high *C. parapsilosis* incidence in pediatric patients [19]. Our study identified only one endemic genotype, which included a cluster of three patients. These patients had been admitted to the hospital at different times (with an interval of 1 year; Table 1); however, the presence of persistent endemic genotypes that adapted to survival in hospital settings is possible. Two of the patients were admitted to the cardiac intensive care unit during the CI event, but the third was not admitted to this care unit. Thus, it is important to mention that the presence of a cluster involving patients who were not geographically and temporally related may be due to limitations of the genotyping procedure [17]. Our findings differ from the previous reports showing clonal occurrence of *C. parapsilosis* sensu stricto in adults during COVID-19 pandemic in Brazilian hospitals [11,14]. The large number of critically ill adult patients with COVID-19 and the overload of the health system could lead to a failure in the strategies of infection control strategies.

This study has some limitations. We did not determine the potential infection source(s) or transmission route(s) in our hospital. Due to the retrospective nature of the study, we had no isolates from the gut microbiota of patients, hospital environment, or healthcare workers’ hands. In this sense, the single complementary analysis performed in this study was the survey of patients with blood cultures with central venous catheter (CVC) tip cultures. Importantly, despite CVC tip cultures being useful in evaluating patients with possible catheter-related bloodstream infection, these results can be misleading, particularly in the absence of concomitant blood cultures [37]. From a total of 51 blood cultures with CVC tip cultures, 23 positive CVC tip cultures were identified, of which 19 showed positive CVC tip with blood culture taxon match, 11 of which were *C. parapsilosis* complex (Appendix A). Four *C. parapsilosis* sensu stricto isolates (from blood and CVC tip culture) of two patients were genotyped by polymorphic microsatellite markers.

Although the genotypes identified from these two patient’s blood samples differed (disregarding a possible horizontal transmission), the two catheter tips isolates had the same genotype as that identified from the blood culture samples, suggesting a catheter-related bloodstream infection in both patients, reinforcing that catheter care is important for source control.

*C. parapsilosis* complex azole resistance rates of > 10% have been reported across multiple regions [8]. Recently, the emergence of FRCP healthcare-associated infections has been increasingly reported in Brazil, some of which showed evidence of horizontal transmission [11,14,16]. Periodic surveillance is needed to identify potential changes in the susceptibility profile of *Candida* species to fluconazole given the increase in fluconazole use and the emergence of FRCP isolates in Brazil with the Y132F point mutation in the *ERG11* gene [38,39]. All *C. parapsilosis* sensu stricto identified in this study (one isolate per IC case) were sensitive to amphotericin B, and one was non-sensitive to fluconazole. Fluconazole resistance may involve point mutations in genes involved in the sterol biosynthesis pathway (especially *ERG11*), overexpression of efflux pumps (*CDR1* and *MDR1*), and mutations in the transcription factors *UPC2*, *TAC1*, and *MRR1*. The Y132F point mutation was the major one detected in FRCP isolates and appears to confer dissemination potential [30,38,40]. This hotspot mutation was not identified in our single isolate that was non-sensitive to fluconazole; instead, we identified a homozygous silent mutation (T591C), which has been previously described in *C. parapsilosis* sensu stricto fluconazole-sensitive and -resistant isolates, therefore not justifying the identified phenotype; this isolate was also identified as weak biofilm producer [16,30,40].

The emergence of fluconazole-resistant species may be related to prior fluconazole use and emphasizes the importance of effectively managing *Candida* infections, including strategies for infection control [38]. Although fluconazole is inexpensive, has limited toxicity, and is available for oral administration, echinocandins have been recommended as a first-line treatment for candidemia by major clinical guidelines [41,42,43]. In the case identified in our study, the patient had no history of fluconazole prophylaxis, and micafungin was administered, which achieved clinical and microbiological cure.

## 5. Conclusions

*C. parapsilosis* sensu stricto is an opportunistic pathogen known to cause hospital outbreaks. Infections are primarily associated with the presence of endemic genotypes, while clonal transmission involving FRCP strains emphasizes the importance of effective management of these infections. However, previous studies have focused on adult patients, while the epidemiology and outcome of IC differ between pediatric and adult patients. Continuous changes in pediatric IC epidemiology have been reported; therefore, epidemiological surveillance is another essential measure for the control and prevention of infections and hospital outbreaks. In the pediatric scenarios evaluated in this study, no clear evidence of clonal outbreaks or emergence of fluconazole-resistant isolates (ERG11-Y132F) were found, suggesting the involvement of multiple sources.

## Figures and Tables

**Figure 1 jof-08-01280-f001:**
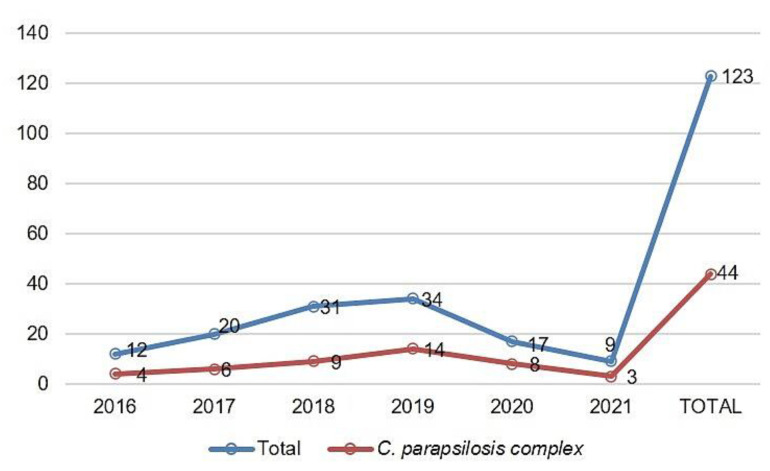
Distribution of *C. parapsilosis* complex in relation to the total of clinical isolates from IC cases per year of the study (August 2016 to August 2021).

**Figure 2 jof-08-01280-f002:**
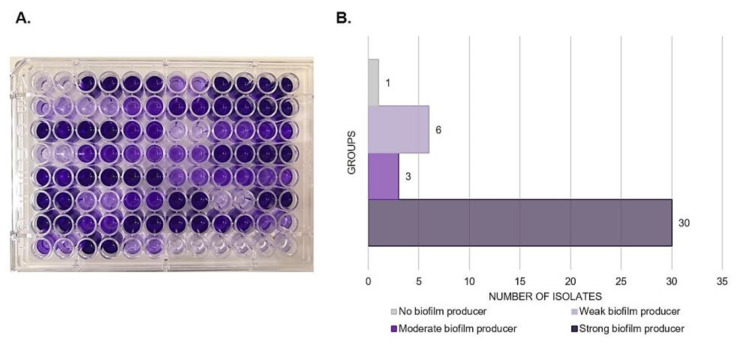
Total biofilm biomass quantified by crystal violet staining in 96-well microtiter plate (**A**). Distribution of *C. parapsilosis* sensu stricto isolates by biofilm production groups (**B**).

**Figure 3 jof-08-01280-f003:**
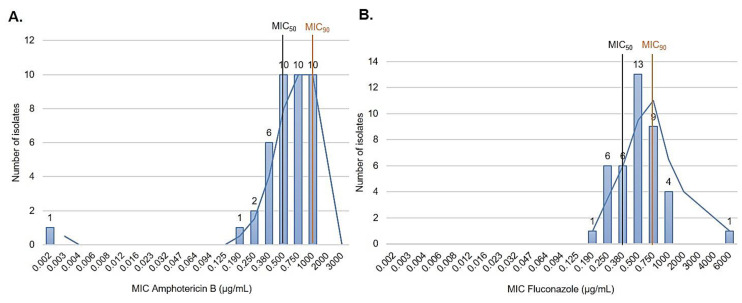
In vitro antifungal susceptibility to amphotericin B (0.002–32 μg/mL) and fluconazole (0.016–256 μg/mL) performed using gradient diffusion strips. (**A**) Amphotericin B and fluconazole (**B**) MIC distribution of 40 *C. parapsilosis* sensu stricto isolates.

**Figure 4 jof-08-01280-f004:**
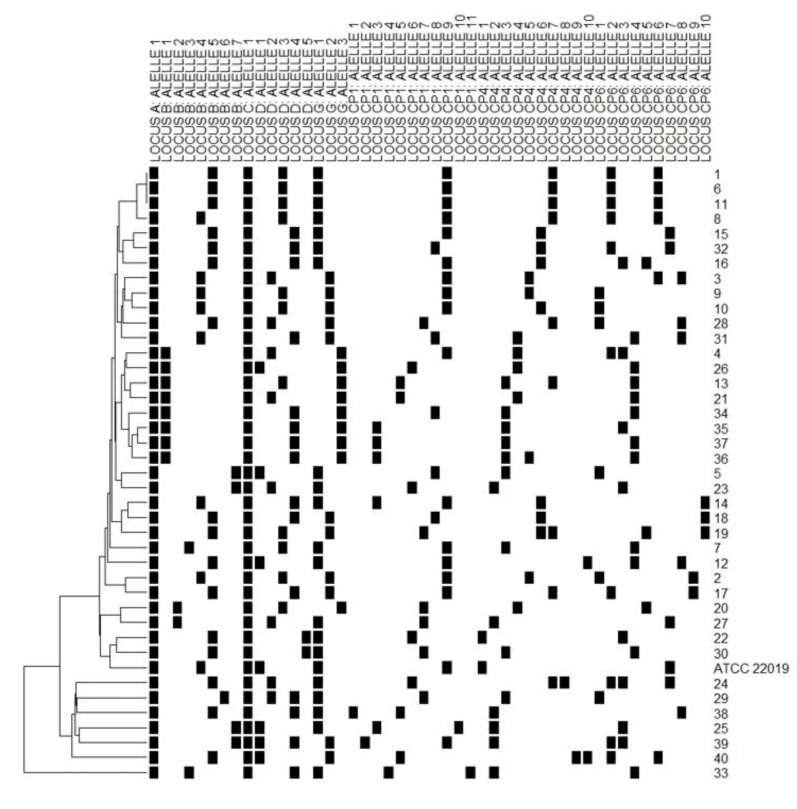
Allelic profile of 40 *C. parapsilosis* sensu stricto isolates and ATCC 22,019, with a dendrogram showing their clustering. Each line indicates the (id) number of isolates. The black square indicates the presence of an amplification product. Each column corresponds to an allele of the respective polymorphic microsatellite marker (A, B, C, D, G, CP1, CP4, and CP6). Isolates 1, 6 and 11 showed the same allelic profile (clones).

**Figure 5 jof-08-01280-f005:**
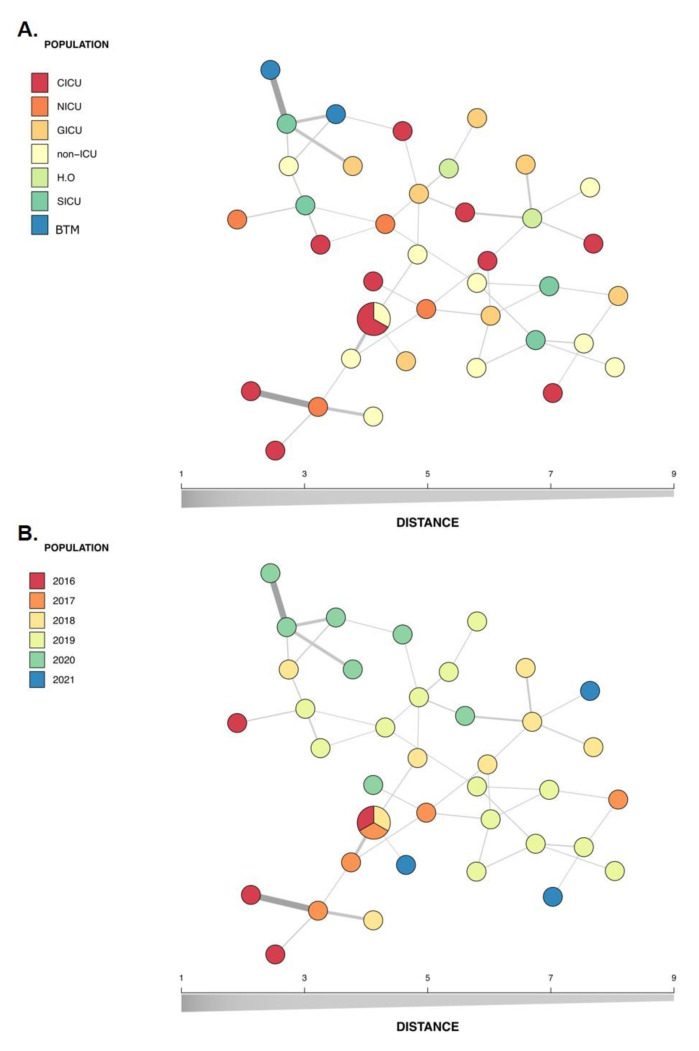
Minimum spanning tree of the relationship between the 38 genotypes among the 40 isolates of *C. parapsilosis* sensu stricto isolated from the bloodstream of infected patients over a 5-year period in a pediatric tertiary hospital. Each circle represents a unique genotype, while circle size corresponds to the number of isolates of the specific genotype and the color of the isolates represent the hospital setting (**A**) or year of isolation (**B**). CICU, cardiac intensive care unit; ICU, intensive care unit; NICU, neonatal intensive care unit; GICU, general intensive care unit; SICU, surgical intensive care unit; HO, hematology-oncology; BMT, bone marrow transplant.

**Table 1 jof-08-01280-t001:** General data on the origin, year of isolation, and microbiologic characteristics of clonal *C. parapsilosis* sensu stricto isolates.

Number	Isolation Date	Hospital Setting	Clinical Sample	Biofilm Assay	Antifungal Susceptibility Test		Outcome
					Amphotericin B (µg/mL)	Fluconazole (µg/mL)	
1	18/08/2016	CICU	Blood	Strong	0.38	0.75	Death
6	02/05/2017	CICU	Blood	Strong	0.50	0.25	Survival
11	02/02/2018	Other non-ICU	Blood	Strong	0.75	0.50	Survival

CICU, cardiac intensive care unit.

## Data Availability

All data that support the conclusions of this study are available upon a reasonable requires from the corresponding author.

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
