# Peer review of "Genotypic Diversity of Candida parapsilosis Complex in Invasive Candidiasis at a Pediatric Tertiary Hospital: A 5-Year Retrospective Study"

_jof, 2022, doi:10.3390/jof8121280_

Round 1

Reviewer 1 Report

This study was a 5-year retrospective analysis of IC cases in a Brazilian tertiary pediatric hospital. The genotyping was performed by Microsatellite typing for C. parapsilosis sensu stricto and they identified 24 high genotypic diversity among the isolates. Antifungal susceptibility was done for two classes of antifungal drugs and I think it would add much more value if the authors included echinocandin drugs, especially micafungin as well. Biofilm formation was also evaluated but the authors didn't clarify if the non-biofilm producer isolate is the one with a silent mutation in the ERG11 gene. 

This study was a 5-year retrospective analysis of 123 IC cases in a Brazilian tertiary pediatric hospital which is sufficient in terms of period and number of isolates for an epidemiology study. There is some missing information to be added to substantiate the findings. The study is well-designed and provides sufficient information regarding the epidemiology of Candida species in the pediatric ward. The introduction and material method parts were well written; however, the result and discussion parts should be extensively revised and clarify the missing information. In total, the manuscript needs to address the missing information and be revised comprehensively to satisfy the requirements.

 The genotyping was performed by Microsatellite typing for C. parapsilosis sensu stricto non-duplicate and they identified high genotypic diversity among the isolates. Many studies have reported the clonal outbreak due to Candida parapsilosis in Brazil, especially between 2020 and 2021 (pandemic period), however, the authors claimed that there was no evidence of clonal outbreaks. Please rationally discuss this conflict.

In the same line, please explain the issue that why the incidence of fluconazole resistance was low in your collection. Is this because of the administration of the non-fluconazole drug in your hospital? Or strict infection control strategies, etc.?

Morpheus heatmap (Figure 3) should be replaced with a phylogenetic tree showing the genotype similarity which is different from the minimum spanning tree.

Line 86, so many cases have reported the evolution of drug resistance in the second sample isolated from the same patient. This could be a reason that you did not find any Candida parapsilosis resistance isolates in your collection. This should be mentioned as a limitation and authors are encouraged to include the second isolates, where available to precisely report resistance prevalence.

Antifungal susceptibility testing was performed for two classes of antifungal drugs, and I think it would add much more value if the authors included echinocandin drugs, especially micafungin as well.

Biofilm formation was also evaluated but the authors didn't clarify if the non-biofilm producer isolate is the one with a silent mutation in the ERG11 gene. Please add a biofilm quantification figure to highlight the high biofilm production of Candida parapsilosis isolates.

Author Response

Response letter

Reviewer #1:

  1. Thanks for your constructive suggestions. We have modified the manuscript and answered all commentaries.

This study was a 5-year retrospective analysis of IC cases in a Brazilian tertiary pediatric hospital. The genotyping was performed by Microsatellite typing for C. parapsilosis sensu stricto and they identified 24 high genotypic diversity among the isolates. Antifungal susceptibility was done for two classes of antifungal drugs and I think it would add much more value if the authors included echinocandin drugs, especially micafungin as well. Biofilm formation was also evaluated but the authors didn't clarify if the non-biofilm producer isolate is the one with a silent mutation in the ERG11 gene. 

  1. Thank you for your suggestions. We agree with you, however the complete susceptibility profile of Candida species isolated at the hospital, by the microdilution method, will be presented in a further study. The main objective of this project was to eveluate the way of dissemination of Candida parapsilosis sensu stricto in this hospital, due to the increasing numbers observed in the last few years. As we had access to fluconazole and amphotericin B gradient diffusion strips, and considering the emergence of resistance to fluconazole, we have included these results in this article.

According to your recommendation we have included a figure as an exemple of the biofilm producing data (figure 2).

This study was a 5-year retrospective analysis of 123 IC cases in a Brazilian tertiary pediatric hospital which is sufficient in terms of period and number of isolates for an epidemiology study. There is some missing information to be added to substantiate the findings. The study is well-designed and provides sufficient information regarding the epidemiology of Candida species in the pediatric ward. The introduction and material method parts were well written; however, the result and discussion parts should be extensively revised and clarify the missing information. In total, the manuscript needs to address the missing information and be revised comprehensively to satisfy the requirements.

The genotyping was performed by Microsatellite typing for C. parapsilosis sensu stricto non-duplicate and they identified high genotypic diversity among the isolates. Many studies have reported the clonal outbreak due to Candida parapsilosis in Brazil, especially between 2020 and 2021 (pandemic period), however, the authors claimed that there was no evidence of clonal outbreaks. Please rationally discuss this conflict.

In the same line, please explain the issue that why the incidence of fluconazole resistance was low in your collection. Is this because of the administration of the non-fluconazole drug in your hospital? Or strict infection control strategies, etc.?

  1. The explanation for this finding is that many of the previous Brazilian studies were in adult population. During the COVID-19 pandemic this group was more affected then the pediatric one. In addition, the large number of critically ill adult patients with COVID-19 and the overload of the health system could lead to a failure in hospital hygiene practices. Also, an increasing use of broad-spectrum drugs and antifungals prophylaxis, both risk factors for IC and resistance emergence, could contribute for the observed diferences between adults and pediatric patients.

Morpheus heatmap (Figure 3) should be replaced with a phylogenetic tree showing the genotype similarity which is different from the minimum spanning tree.

  1. According to your recommendation we have replaced the figure (figure 4).

Line 86, so many cases have reported the evolution of drug resistance in the second sample isolated from the same patient. This could be a reason that you did not find any Candida parapsilosis resistance isolates in your collection. This should be mentioned as a limitation and authors are encouraged to include the second isolates, where available to precisely report resistance prevalence.

  1. We accept your suggestion and, at this point in the discussion, we reinforce that only one isolate per case of IC was analyzed. In order to avoid overestimate the IC data and impair the clonality analysis, we chose to exclude the second isolate from the stydy. We believe that the inclusion of a second isolate would be justified in a context of previous use of azole or therapeutic failure, but this data has not yet been evaluated.

Antifungal susceptibility testing was performed for two classes of antifungal drugs, and I think it would add much more value if the authors included echinocandin drugs, especially micafungin as well.

  1. Thank you. We agree with your suggestion, however the complete study with susceptibility profile data of all Candida species will be presented separately.

Biofilm formation was also evaluated but the authors didn't clarify if the non-biofilm producer isolate is the one with a silent mutation in the ERG11 gene. Please add a biofilm quantification figure to highlight the high biofilm production of Candida parapsilosis isolates.

  1. Thank you. We agree with your suggestion, we have included this information in the manuscript.

Reviewer 2 Report

In my opinion, the work submitted for review is interesting and will be of interest among readers. The results are unique and are a good complement to the current literature. The manuscript is carefully written and its individual chapters are cohesive. Although I have minor remarks regarding the graphic side of the work. Please improve the Figs quality.

Author Response

Response letter

Reviewer #2:

  1. Thanks for your constructive suggestion.

In my opinion, the work submitted for review is interesting and will be of interest among readers. The results are unique and are a good complement to the current literature. The manuscript is carefully written and its individual chapters are cohesive. Although I have minor remarks regarding the graphic side of the work. Please improve the Figs quality.

  1. Thank you for reviewing the manuscript, we improved the figures quality.
